# Update on Robotic Total Mesorectal Excision for Rectal Cancer

**DOI:** 10.3390/jpm11090900

**Published:** 2021-09-08

**Authors:** Simona Giuratrabocchetta, Giampaolo Formisano, Adelona Salaj, Enrico Opocher, Luca Ferraro, Francesco Toti, Paolo Pietro Bianchi

**Affiliations:** 1Division of General and Robotic Surgery, Dipartimento di Scienze della Salute, Università di Milano, ASST Santi Paolo e Carlo, 20142 Milano, Italy; giampaolo.formisano@asst-santipaolocarlo.it (G.F.); adelona.salaj@asst-santipaolocarlo.it (A.S.); luca.ferraro@asst-santipaolocarlo.it (L.F.); francesco.toti@asst-santipaolocarlo.it (F.T.); PaoloPietro.Bianchi@unimi.it (P.P.B.); 2Division of General and HPB Surgery, Dipartimento di Scienze della Salute, Università di Milano, ASST Santi Paolo e Carlo, 20142 Milano, Italy; enrico.opocher@unimi.it

**Keywords:** robotic surgery, rectal cancer, total mesorectal excision, robotic low anterior resection

## Abstract

The minimally invasive treatment of rectal cancer with Total Mesorectal Excision is a complex and challenging procedure due to technical and anatomical issues which could impair postoperative, oncological and functional outcomes, especially in a defined subgroup of patients. The results from recent randomized controlled trials comparing laparoscopic versus open surgery are still conflicting and trans-anal bottom-up approaches have recently been developed. Robotic surgery represents the latest consistent innovation in the field of minimally invasive surgery that may potentially overcome the technical limitations of conventional laparoscopy thanks to an enhanced dexterity, especially in deep narrow operative fields such as the pelvis. Results from population-based multicenter studies have shown the potential advantages of robotic surgery when compared to its laparoscopic counterpart in terms of reduced conversions, complication rates and length of stay. Costs, often advocated as one of the main drawbacks of robotic surgery, should be thoroughly evaluated including both the direct and indirect costs, with the latter having the potential of counterbalancing the excess of expenditure directly related to the purchase and maintenance of robotic equipment. Further prospectively maintained or randomized data are still required to better delineate the advantages of the robotic platform, especially in the subset of most complex and technically challenging patients from both an anatomical and oncological standpoint.

## 1. Introduction

Surgery for rectal cancer remains challenging due to its critical anatomical, oncological and technical issues. Total Mesorectal Excision (TME), first described by Heald in 1982 [1], is the gold standard for the treatment of rectal cancer; a precise dissection along an avascular embryologically based plane is performed in order to achieve good oncological and functional results. However, a debate still exists on the best surgical approach for TME, especially in challenging patients (i.e., low-lying tumors, high BMI, prior chemoradiotherapy and unfavorable pelvimetry) and the trans-anal approach has been described over the last years to overcome the limitations of conventional laparoscopy [2,3,4,5].

Laparoscopic surgery for colon cancer can achieve comparable oncological outcomes and superior short-term postoperative outcomes when compared to open surgery [6,7,8]. As far as rectal cancer is concerning, some randomized controlled trials have validated laparoscopic surgery to be as safe and effective oncologically as the open approach, with better postoperative outcomes [9,10,11,12,13,14], while others have failed to prove its non-inferiority from an oncological standpoint [15,16], even at a 2-year follow up [17]. These data show that laparoscopic TME still remains technically challenging, due to the technical limitations of conventional laparoscopy. Moreover, the complexity of TME is mainly increased in patients with a narrow pelvis, male and/or obese patients and patients with bulky tumors and all these factors could potentially negatively affect operative time, specimen quality, complication rate and, ultimately, survival.

Robotic surgery may potentially overcome these drawbacks thanks to its 3D high-definition vision, the use of a stable camera and the wrist-like movement of the instruments allowing seven degrees of freedom and tremor filtration. The wristed tip of the robotic instrument allows for correct triangulation, traction and countertraction even deep in the pelvis for optimal tissue handling. The first report about robotic TME was in 2006 by Pigazzi et al. [18]. Since then, several papers comparing robotic rectal surgery (RRS)/TME with laparoscopic rectal surgery (LRS) and open rectal surgery (ORS) have been published. Although the largest and only available Randomized Controlled Trial (RCT) published to date [19] failed to demonstrate superiority in the conversion rate for robotic compared to laparoscopic rectal surgery, robotic surgery may have a positive impact in most challenging cases [19] and several population-based studies have shown benefits in terms of conversions, complications, functional outcomes and length of hospital stay [20,21,22,23,24,25,26,27,28,29]. This paper aims to describe the surgical technique of robotic TME and review the recent literature on robotic rectal surgery, focusing on short-term, oncological, functional outcomes and costs.

## 2. Surgical Technique

The procedure is a full-robotic TME, performed with the Davinci Xi Surgical Platform (Intuitive Surgical Inc., Sunnyvale, CA, USA). Compared to previous robotic platforms, this system allows for an easier setup and multiquadrant access, faster docking and simple OR setup (boom-mounted rotating arms).

### 2.1. Patient Positioning and Docking

The patient is placed supine in a modified lithotomy position with arms alongside the body and abducted legs positioned in adjustable stirrups. The patient is carefully secured with a dedicated patient soft foam pad (Pink Pad, Xodus Medical Inc., New Kensington, PA, USA) to prevent sliding in steep Trendelemburg. The first assistant stands on the patient’s right side and the cart is placed at the patient’s left side, docked from the left lower quadrant over the left hip (Figure 1). The patient is then placed in a 20–25° Trendelemburg position with a 20–25° right tilt (Figure 2).

A Verres needle is inserted in the left hypochondrium (Palmer’s point) for the induction of 12 mmHg pneumoperitoneum. A 12 mm optical port is inserted in the right flank. Four 8 mm robotic trocars are then inserted along a straight line that is parallel to and about 4 cm cranial to the costofemoral line; a distance of 6–8 cm between each port is maintained. An additional 8 mm robotic port is placed in the left flank and will be used for the TME and vascular control. A 5 mm laparoscopic port may be introduced in the right hypochondrium to optimize the assistant’s tractions, if required (Figure 3).

The cart is deployed for docking from the patient’s left side. In order to define the correct cart position, a green laser is emitted from the overhead boom to the optical port and the guided docking procedure is completed.

### 2.2. Step-By-Step Technique

A tip-up fenestrated grasper, Bipolar forceps and a permanent cautery hook (or Monopolar curved scissors) are mounted in R1, R2 and R4, respectively.

Full-robotic TME is essentially based on three steps:-Vascular control;-TME;-Splenic flexure mobilization.

### 2.3. Vascular Control

The sigmoid colon is lifted up with the robotic grasper in R1 (8 mm robotic trocar in the left flank). The peritoneum is then incised at the level of the sacral promontory to access the avascular presacral mesorectal plane, where the hypogastric nerves are identified and preserved and a medial to lateral dissection is performed to identify the left ureter and gonadal vessels. The superior rectal artery is identified as a landmark and the dissection continues in a bottom-to-up fashion. The origin of inferior mesenteric artery (IMA) is thus identified with the surrounding lymphatic tissue, dissected at its origin and divided with Hem-o-lok^®^ clips.

The medial to lateral dissection is performed underneath the inferior mesenteric vein (IMV) and the Toldt–Gerota plane is identified as a landmark.

The IMV is dissected, isolated and transected using Hem-o-lok^®^ clips (Teleflex, Wayne, PA, USA). The dissection continues downward in a medial-to-lateral fashion, with the R1-grasper lifting the descending-sigmoid mesocolon and the assistant surgeon providing countertraction on the Gerota’s fascia. Coloparietal detachment is then completed along the white line of Toldt.

### 2.4. TME

TME is carried out according to the Heald’s embryologically based principles along the avascular plane in order to preserve the hypogastric nerve and the sacral venous plexus. Frequent repositioning of the grasper in R1 is fundamental to maintain the right countertraction for dissection.

The dissection starts posteriorly along the plane between the endopelvic visceral fascia and endopelvic parietal fascia. R1 is used to provide upward traction on the mesorectum, with the wrist joint in a L-shape to allow for a larger area of retraction; the assistant maintains a cranial traction on the sigmoid colon and R2 and R4 are the operative arms, with a bipolar grasper and a monopolar hook. The right lateral and the anterior plane are then dissected up to the seminal vesicles in a counterclockwise fashion; R1 is now used to lift the peritoneum of the Douglas pouch and vagina and the seminal vesicles and prostate are freed and protected. The left lateral pelvic fascia is then dissected up to its lower portion until the pelvic nerve plexus is identified and the “bare rectum area” is visualized. The mesorectal dissection is performed in a cylindric fashion until past the level of the lesion and the access to the levator ani plane is gained (Figure 4). During this phase in the lower mesorectum, a 0° camera may be helpful to achieve a better visualization. Rectal transection is performed with a robotic stapler after the evaluation of the vascular perfusion of the rectal stump through the integrated fluorescence imaging system. Stapled end-to-end low/ultralow colorectal anastomosis or manual coloanal anastomosis are performed according to the tumor distance from the anal verge. A transabdominal robotic top-down targeted transection of the levator ani plane may be performed if an abdominoperineal/extralevator abdominoperineal excision is required.

### 2.5. Splenic Flexure Mobilization

Splenic flexure mobilization is performed in the vast majority of the patients to provide for tension-free colorectal anastomosis, with a few exceptions in the case of a very long and redundant sigmoid colon. During this step of the procedure, the robotic grasper in R1 is moved to the epigastric trocar.

Splenic flexure takedown is performed with a one-inch one-inch bottom-up approach.

The Toldt–Gerota plane previously developed and the IMV are identified. The transverse colon is lifted up with a R1-grasper and the lesser sac is opened through the incision of the transverse mesocolic root at the level of the anterior pancreatic border, gaining access to the lesser sac (one-inch one-inch bottom-up approach) [30,31] (Figure 5). The assistant keeps holding the transverse colon and the R1 grabs the posterior side of the stomach to achieve optimal exposure. A medial-to-lateral approach is carried out along the pancreatic body. The splenic flexure is then retracted medially by the assistant and the mobilization is completed from the inferior splenic pole to the previous plane along to the white line of Toldt. Coloparietal and coloepiploic detachment are performed and splenic flexure takedown is thus completed.

Different approaches have been described for splenic flexure takedown, depending on the surgeons’ preference, with different trocar layouts and table tilting (i.e., reverse Trendelemburg). Some authors are used to performing this step laparoscopically; thus, the robot is docked for the core of the procedure, namely TME [32].

Regardless of the surgical approach (robotic or laparoscopic), four ways to mobilize the splenic flexure have been described [33]. The medial approach (medial-to-lateral) involves an extensive dissection of the medial plane separating the descending mesocolon from the Toldt fascia; the sovramesocolic approach (top-to-bottom) starts with a gastrocolic ligament transection to enter the lesser sac; the lateral approach begins with coloparietal detachment along the white line of Toldt; finally, the “one inch-one inch” approach allows access to the lesser sac in a bottom-to-up fashion through the transection of the transverse mesocolic root along the pancreatic fusion fascia.

In our experience, a full-robotic low anterior resection is performed. Splenic flexure takedown can be challenging and the robotic approach could represent a valuable tool in facilitating this step of the procedure when compared to conventional laparoscopy. Moreover, the new robotic platform (DaVinci Xi system) is capable of extensive multiquadrant access and significantly reduces external arm collisions and docking time and allows the procedure to be performed without any change in table tilting/docking (steep Trendelenburg position with right tilt for splenic flexure takedown, vascular control and TME).

The bottom-up approach, with stable exposure and traction exerted by the third arm on the transverse mesocolon, allows for easy access into the lesser sac and dissection along embryological planes (pancreatic fusion fascia) over the body and tail of the pancreas, thus preserving the integrity of the proper mesocolic fascia and its blood supply as well.

## 3. Discussion and Literature Review

The potential advantages of RRS compared to conventional LRS have been widely discussed in many studies with different levels of evidence. Due to the limited space and maneuverability of instruments and the influence of tremor fulcrum effects as well, performing laparoscopic surgery in a narrow pelvis could be challenging. Technical improvements in robotic instruments and technology may provide advantages in vision, ergonomics, dexterity and wristed articulation, leading to a better surgical performance and consequently potential better short-term, oncological, functional outcomes as well.

### 3.1. Intraoperative and Short-Term Postoperative Outcomes

Data from the meta-analysis and RCTs reported a longer operative time for the robotic approach compared to laparoscopy and open surgery [19,20,21,22,23,24,25,26,27,34,35,36,37,38,39,40,41,42,43]. This was mainly attributed to time-consuming double-docking procedures and the changing of the robotic instruments. Over the last few years, the increased use of the DaVinci Xi platform with its technology improvements seems to be associated with a significant reduction in operative time [28,44], which is almost comparable to laparoscopy. This could be mainly due both to the improvements in the technology itself (endoscope inserted in any arm, multiquadrant access, longer instruments), leading to a significantly decreased docking time and instrument switch and the improvement in the surgeon’s learning curve that flattens with time and experience.

As far as the conversion rate is concerned, the ROLARR trial [19] failed to demonstrate superiority in the conversion rate for RRS compared to LRS. However, meta-analysis [35,45], many population-based studies and a national database [20,21,22,23,24,25,26] including thousands of patients showed a significantly lower conversion rate in the robotic group compared to laparoscopy, especially in rectal surgery and in the high-risk patients subgroup (male, neoadjuvant radiochemiotherapy, T3N1 patients) [20,21,46], probably reflecting what actually happens in daily practice. Those data have been confirmed by a recent large retrospective cohort study and a logistic regression analysis involving 600 patients [47,48] showing that the type of surgery (laparoscopic vs. robotic approach) and obese patients are independent risk factors for conversion.

Although no significant statistical difference has been seen in the complication rate among robotic, laparoscopic and open groups in some studies [28,35,36,45,49,50], other papers have reported data trending in favor of robotic surgery. The ROLARR trial reported a similar complication rate in both the robotic and laparoscopic group, 14% vs. 16.5%, respectively, related in particular to anastomotic leakage, 9.9% and 12.2%, respectively [19]. In a recent meta-analysis, Simillis et al. [27] reported lower wound infections due to shorter skin incisions and less contamination. The same results have been reported in large population-based studies, including about 11,000 patients with a lower overall septic complication rate (1.6% in robot vs. 3.1% in lap, *p* value = 0.02) and a lower wound dehiscence rate (0.1% in robot vs. 0.7% in lap, *p* value = 0.05) in the robotic group [21,23,24]. As complications increase, postoperative morbidity and mortality proportionally increase as well, mostly in low-volume hospitals and if surgery is performed by a less trained surgeon [46].

The aforementioned population-based studies [23,24,25,26] also reported a significant reduction in the length of stay in favor of the robotic group (3.8–4.8 vs. 4.7–6.3 days, *p* < 0.001, robotic vs. laparoscopic group, respectively), that is probably strictly related to the reduction in the conversion and complication rates [31]. The same figures have been reported as far as a shorter time to first flatus is concerned [27].

### 3.2. Functional Outcomes

Functional outcomes are among the most important issues related to pelvic surgery since they could ultimately impact and affect patients’ postoperative quality of life. This topic has been widely investigated by comparing the functional outcomes in the RRS and LRS groups, based on the assumption that a better 3D visualization of anatomical structures and more precise movements may enable the surgeon to preserve the autonomic nerves and function. Although most reports are from case-series, with different measurements, follow-ups and small sample sizes, two recent meta-analyses [51,52] reported better functional results after robotic surgery for rectal cancer when compared to conventional laparoscopy. Regarding urinary function in men at 6 months after surgery, the IPSS (International Prostatic Symptoms Score) was significantly improved in the RRS group compared to the LRS group and these data were confirmed at a 12-month follow-up. IIEF (International Index of Erectile Function) scores were significantly in favor of the RRS group at 6- and 12-months post-operation. Mixed urinary and sexual function outcomes were also reported for women in many case series, with no significant differences in meta-analysis results. The available evidence of the potential functional benefits of robotic surgery over traditional laparoscopy after rectal cancer resection should be confirmed by high-quality randomized or prospective multicenter studies adequately powered for functional outcomes and patient-reported outcomes and quality of life.

### 3.3. Oncological Outcomes

Circumferential resection margin (CRM), distal resection margin (DRM) and the mesorectal grading system according to the Quirke criteria [53] are universally considered as valuable surrogates for long-term local recurrence rates and oncological outcomes as shown by several studies [10,54,55,56,57,58,59].

The ROLARR Trial reported a CRM positivity of 5.1% and 6.3% in the robotic and laparoscopic group, respectively, with no statistically significant difference between the two groups. DRM involvement and the pathological assessment of the quality of the plane of surgery were also comparable [19]. Another RCT from Korea [43] reported the same figures, with no statistically significant difference between robotic and laparoscopic surgery as far as pathological outcomes are concerned.

A recent metanalysis [58] has shown that robotic surgery is the better way to achieve a complete TME. Nevertheless, the lack of quality and heterogeneity among the included studies must be underlined. Based on the quality of the available evidence and according to pathological outcomes as surrogates for long-term oncological results, it cannot be concluded that robotic surgery is actually superior to its laparoscopic counterpart.

The survival data from the ROLARR trial are still unavailable and reports of long-term oncologic outcomes for robotic rectal surgery remain limited due to the relatively recent uptake of robotic surgery. Park et al. [59] found no differences in the 5-year OS, DSF and LR rates. Similar results were reported by Cho et al. in a case matched series of 278 patients [60], with a 5-year OS of 92.2% and a DFS of 81.8% in the robotic group. More recently, Kim et al. [61] showed that robotic surgery was a significant positive prognostic factor for OS and cancer-specific survival in a multivariate analysis. Recently, Park et al., in their retrospective single-center propensity score-matched analysis focusing only on mid-low rectal cancers, reported that robotic surgery had similar overall 5-year survival figures when compared to laparoscopic surgery. However, they found robotic surgery to be beneficial in a subgroup of patients who received preoperative chemoradiation and had ypT3–4 tumors after neoadjuvant treatment. The 5-year distant and local recurrence rates were 44.8% and 5.0% in the laparoscopic group and 9.8% and 9.8% in the robotic group, respectively and reached statistical significance. These data suggest that robotic surgery may be beneficial in most complex cases with high-risk features of recurrence [62].

The ongoing European RESET trial [63] (prospective, observational, case-matched, four-cohort patients) has been designed to analyze TME outcomes with open, laparoscopic, robotic and trans-anal approaches in a complex subset of patients from both an anatomical and oncological standpoint (distance from the anal verge, intertuberous distance, obesity and T stage). The preliminary results are still awaited.

### 3.4. Costs

One of the most critical issues for robotic surgery remains the cost, with institutions and payers being concerned about acquisition, maintenance, equipment and implementation. Actually, most of the available studies in different surgical specialties and subspecialties show higher costs related to robotic surgery when compared to its laparoscopic counterpart [32,64,65]. Unfortunately, to date, most of the studies have focused only on the direct costs related to the purchase and maintenance of the robot, along with the related instrumentation per procedure. Although short-term outcomes such as conversions, length of stay and complications seem to be favorable in the robotic surgery groups compared to conventional laparoscopy, these data are rarely taken into account and analyzed as a “total” and inclusive episode cost with a reduced financial burden related to hospitalization, medications and follow-up visits that may reduce the overall costs and counterbalance the excess of expenditure related to robotic equipment. The indirect costs related to the higher conversion rate compared to open surgery, with a prolonged length of stay, postoperative complication management and a delayed return to daily activities are rarely investigated, although they have a significant negative impact on the overall financial burden for each Institution. Cleary et al. [66] in an observational study from a linked data registry including clinical data from the Michigan Surgical Quality Collaborative (2012–2015) specifically showed that the conversion to open surgery significantly increases the payments associated with minimally invasive colorectal surgery. Since conversion rates are lower in robotic versus laparoscopic surgery, the excess expenditures attributable to robotics are attenuated by the consideration of the cost of conversions itself [66,67]. Moreover, as data show that the conversion rate is significantly lower in large volume hospitals, it seems that this variable may impact on indirect costs as well [46]. Therefore, large case series per institution, multidisciplinary team utilization and, ideally, the presence of industry competition are key factors that could reduce the financial burden and make robotic surgery a cost-effective technique.

## 4. Conclusions

Rectal surgery remains challenging, especially in male patients, obese patients and patients with a narrow pelvis and bulking tumors. Robotic technology applied to pelvic surgery may potentially offer clinical and oncological benefits, due to the more precise visualization and dissection along the embryological planes. To date, population-based studies with large sample sizes and metanalyses comparing robotic and laparoscopic surgery have reported better statistically significant results in terms of conversion rates, complications, length of stay and functional outcomes. The excess of expenditures related to robotic surgery may be balanced and mitigated by better short-term results that should be included in future cost analysis studies. Results of prospective multicenter studies focusing on technically challenging patients with high-risk features for recurrence are awaited.

## Figures and Tables

**Figure 1 jpm-11-00900-f001:**
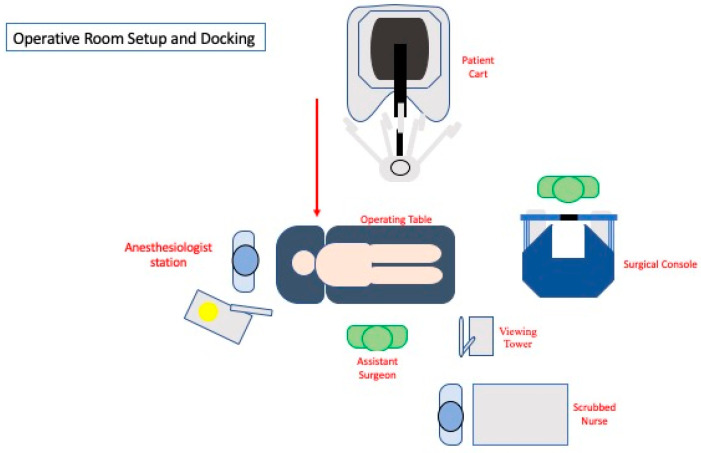
Operative room setup.

**Figure 2 jpm-11-00900-f002:**
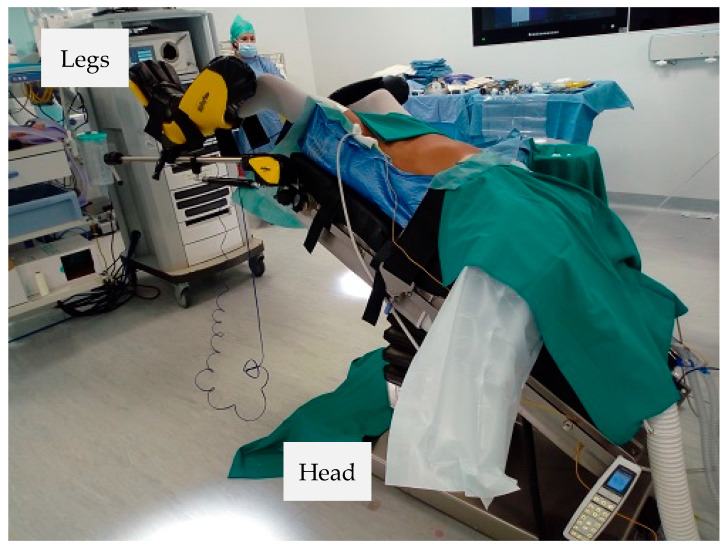
Patient positioning.

**Figure 3 jpm-11-00900-f003:**
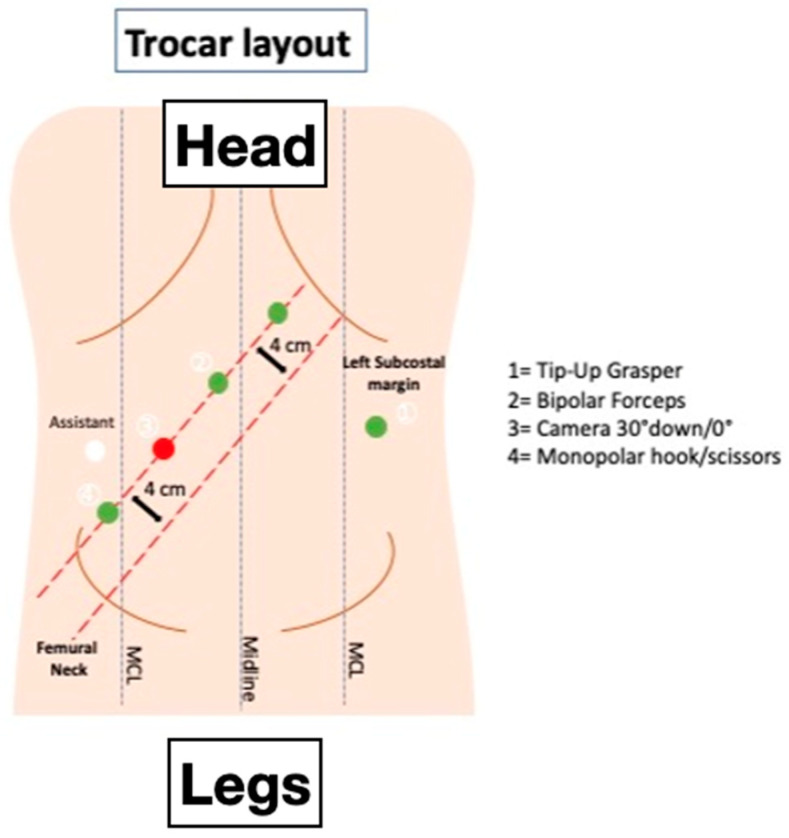
Trocar layout for a fully robotic TME.

**Figure 4 jpm-11-00900-f004:**
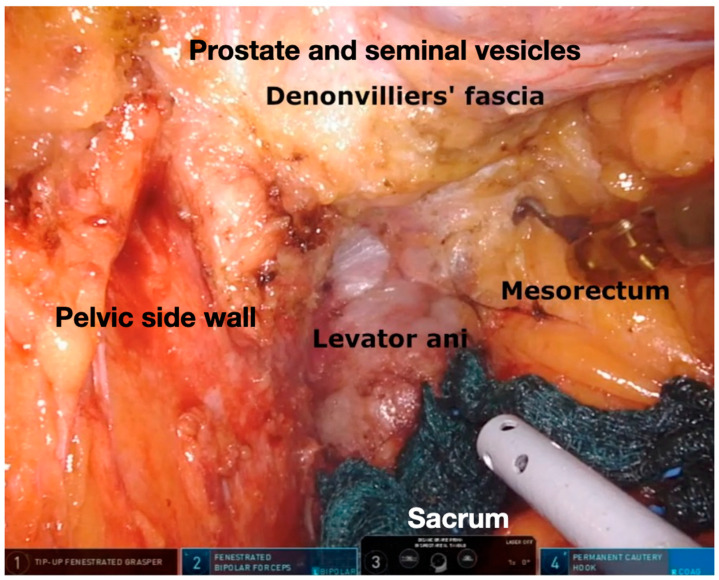
Final view after Total Mesorectal Excision.

**Figure 5 jpm-11-00900-f005:**
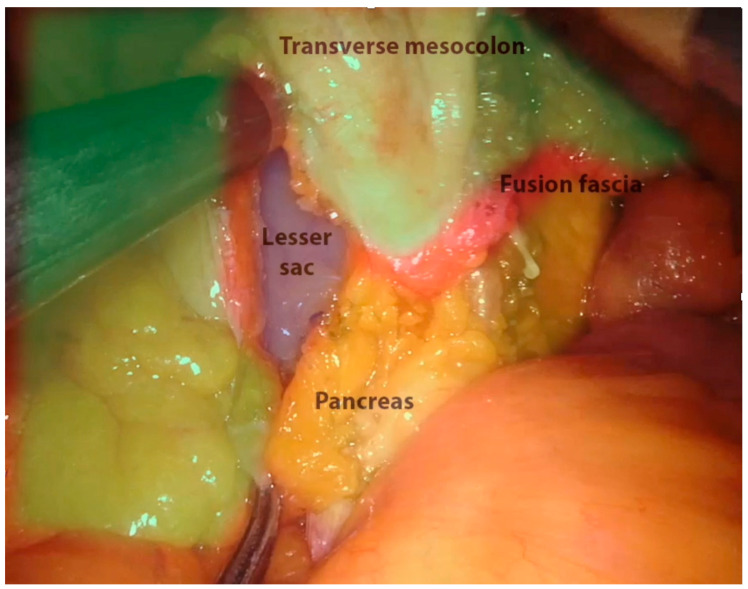
Splenic flexure takedown—**bottom** to **up** approach.

## Data Availability

Not applicable.

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
