# Peer review of "Update on Robotic Total Mesorectal Excision for Rectal Cancer"

_jpm, 2021, doi:10.3390/jpm11090900_

Round 1

Reviewer 1 Report

This paper presents a brief description of the surgical technique using robotic total mesorectal excision, as well as a review of the literature on the use of this technique, its results, and consts. Although the article is interesting, its presentation could be improved. In the review section, the authors focus on comparing laparoscopic and robotic approaches. Therefore, in my opinion, the authors should add more graphics and photos to present the described technique in a more "holistic" way that would reflect the description of the text more comprehensively (not only the results of the operation, but also the actual use of the equipment) to attract the attention of a wider audience than surgeons and readers familiar with medical anatomy. Authors should also pay attention to the abbreviations used, as not all are explained when they first appear in the text.

Author Response

Thanks for the revision. As suggested, more photos have been added regarding the use of robotic equipment (see figure 1,2,3) and intraoperative steps, to improve audience attention and interest. Abbreviations have been revised.

Reviewer 2 Report

The authors have presented a very well structured and comprehensive overview of the current state of knowledge about robotic TME in rectal cancer.
All essential aspects are presented in an understandable way, even for colleagues who are not surgical specialists.
Although this is neither a systematic review nor a meta-analysis, the main current evidence is well and clearly summarized.
Nevertheless, the publication can unfortunately only be rated as an expression of opinion by the working group. No other claim is formulated, but unfortunately the publication cannot be assessed with a high degree of originality and has no significant influence on the current field of research. The current issues identified in the field are known and no relevant new hypotheses are being formulated.
For a non-surgical target group, the images should be labeled more precisely and the spatial orientation should be supplemented. The digits can hardly be identified on the diagram of the trocar positions.
A comparative table with the alternatives to the essential aspects could be helpful for a better overview, but in my opinion it is not mandatory.
Most of the working groups choose the same or a similar operative procedure. The core elements were presented very well.
In addition, there are suitable alternative ways of performing the operation. This applies, for example, to the trocar positions, the preparation routes and the sequence of the operative steps. It is neither sensible nor possible to list all variants, but a general update should at least point out significant variations.

Author Response

Thanks for the revision and for useful suggestions provided.

  1. This paper describes a standardized robotic technique for TME performed in a high-volume colorectal surgery centre. Authors' aim is to describe the step-by-step surgical technique for robotic TME evaluating the actual evidence, without assessing a meta-analysis or systematic review. The robotic TME is a challenging procedure and providing a detailed description could be interesting and useful.
  2. As suggested, some variants to the described surgical technique have been provided and added to the text, as a brief citation. 
  3. Figures have been improved